# Graphitized-rGO/Polyimide Aerogel as the Compressible Thermal Interface Material with Both High in-Plane and through-Plane Thermal Conductivities

**DOI:** 10.3390/ma14092350

**Published:** 2021-04-30

**Authors:** Peng Lv, Haiquan Cheng, Chenglong Ji, Wei Wei

**Affiliations:** Department of Optoelectronic Information Science and Engineering, College of Electronic and Opitical Engineering, Nanjing University of Posts & Telecommunications, Nanjing 210023, China; 1218022710@njupt.edu.cn (H.C.); jichenglong2021@163.com (C.J.); weiwei@njupt.edu.cn (W.W.)

**Keywords:** thermal interface materials, reduced graphene oxide, polyimide, compressibility, thermal conductivity

## Abstract

Reduced graphene oxide (rGO) aerogels with a three-dimensional (3D) interconnected network provides continuous heat transport paths in multi-directions. However, the high porosity of rGO aerogels commonly leads to very low thermal conductivity (TC), and defects and grain boundaries of rGO sheets result in a high extent of phonon scattering, which is far from satisfying the requirement of thermal interface materials (TIMs). Here, a compressible graphitized-rGO/polyimide (g-rGO/PI) aerogel was prepared by the ice-template method and “molecular welding” strategy. The regular cellular structure and closely packed cell walls bring the g-rGO/PI aerogel high compressibility, which made the aerogel can maintain the continuous thermal transport paths well even in highly compacted status. The rGO sheets in the cell wall surface are welded up by g-PI during imidization and graphitization treatment, providing efficient channels for phonon transportation in the 3D network. The g-rGO/PI aerogel in a compressive strain of 95% has a high TC in the plane of 172.5 W m^−1^k^−1^ and a high TC through the plane of 58.1 W m^−1^k^−1^, which is superior to other carbon-based TIMs previously reported.

## 1. Introduction

With the rapid development of portable devices and high-power electronics, severe heat dissipation issues greatly threaten the reliability and performance of the high-tech devices [1,2,3]. To decrease the thermal interface resistance between the heat producing component and the radiator, thermal interface materials (TIMs) are usually inserted between them to solve the interfacial heat transfer problem [4,5]. Graphene, a single layer membrane from sp^2^-hybridized C atoms, possesses a high in-plane thermal conductivity (TC) of up to ~5000 W m^−1^K^−1^. In addition, graphene has high temperature resistance, low thermal expansion and excellent mechanical properties, making it a promising candidate for TIMs [6].

Many efforts have been devoted to constructing macroscale graphene-based monoliths as TIMs. Horizontal graphene papers [7,8,9,10,11,12,13] and vertical graphene monoliths [14,15,16,17,18,19] present ultra-high TC values and have attracted a lot of attention. However, vertical graphene monoliths or horizontal graphene papers only can provide high TC values in a one-dimensional direction. Horizontal graphene papers are commonly used as the heat spreading components due to their high in-plane TC values (1100–3200 W m^−1^K^−1^ [7,8,9,10,11]). However, the through-plane TC values of the horizontal graphene papers are usually low (<10 W m^−1^K^−1^) [11,12,13]. Vertical graphene monoliths present high through-plane TC values (35–680 W m^−1^K^−1^) [14,15,16,17,18] due to the vertical orientation of graphene layers. However, the TC values of the vertical graphene monoliths along the in-plane direction are far lower than that along the through-plane direction, arising from the high anisotropy [19]. The excellent thermal transport performances only along a one-dimensional direction are attributed to the van der Waals forces between graphene layers hindering the transport of phonons leading to thermal resistance [20]. Therefore, these graphene-based TIMs reported in previous literatures [7,8,9,10,11,12,13,14,15,16,17,18] are far from satisfying the requirement in advanced TIMs for portable devices and high-power electronics.

Graphene monoliths (e.g., foam, aerogel and sponge) with three-dimensional (3D) structures and interconnected networks are significantly different from those of vertical graphene monoliths and horizontal graphene papers with highly anisotropic structures [21,22,23]. For these graphene monoliths, the thermal fluxes conduct along the 3D graphene backbone, extending the excellent thermal transport performances of 2D graphene sheets to macro-scale graphene monoliths [24,25,26]. However, 3D graphene monoliths commonly present very low TC values due their ultra-high porosity (>90%). To obtain high-performance TIMs, 3D graphene monoliths should possess high compressibility to be compacted to reduce the porosity. Zhang et al. [24] pressed a graphene foam (thickness of 3 mm) into a thin buckypaper sheets (thickness of around 2–3 μm). The thermal interfacial resistance of the compacted graphene foam is one order of magnitude lower than commercial thermal paste-based TIMs. However, the 3D continuous network of the graphene foam was seriously broken in the ultra-high compressive strain of about 99.9%. Liu et al. [27] infiltrated the bisphenol-A epoxy resin into the graphene aerogels under compression (compressive strain = 70%). The as-prepared graphene aerogel/epoxy composite showed high through-plane TC of 20 W m^−1^K^−1^, which is far higher than that of the composite of the graphene aerogel without compression during the infiltration process (6.51 W m^−1^K^−1^). However, the low intrinsic TC of polymer in the composites prevents the composites achieving higher TC. Our previous work [28] presented an elastic reduced graphene oxide/carbon nanotube (rGO/CNT) aerogel. The hybrid aerogel maintained the continuously heat conduct paths well even under 80% compressive strain. The TC values of the hybrid aerogel were improved significantly by simply mechanical compression. However, there are still some drawbacks for the rGO/CNT aerogel as TIMs: 1. The loose structure of the cell walls leads to high thermal resistance. The compressibility of the aerogel is not high enough to reduce the porosity remarkably.

Recently, the “molecular welding” strategy has been employed to improve the TC of rGO-based TIMs. Yang’s group [29] introduced graphitized polyimide (PI) into the rGO films by in situ polymerization, imidization, and graphitization processes. PI was used as a solder to “weld” the rGO sheets up by the covalent bonding and to fill up the gaps between the rGO sheets. The graphitized-rGO/PI (g-rGO/PI) film presented an enhancement of 21.9% in the in-plane TC as compared to the pure graphitized-rGO (g-rGO) film. They also modified the GO sheets to provide more reactive sites for PI on GO sheets. The in-plane TC of graphitized modified-rGO/PI film was 92.3% higher than that of pure g-rGO film [30]. Zhang et al. [31] welded the junctions in the CNT foam by coating a PI layer followed by a graphitizing treatment to decrease the internally thermal contact resistance of CNT foam. The graphitized PI layers bridge the unconnected CNTs in the CNT foam to form a continuously thermal conductive network in composites.

In this work, a g-rGO/PI aerogel was fabricated by the ice-template method and “molecular welding” strategy to obtain the highly compressible TIMs with high TC values along both the in-plane direction and through-plane direction. During the ice-template assembling process, integrated cell walls and regularly arranged cellular structure were formed, giving the aerogel high compressibility. The aerogel can maintain the interconnected network well at 95% compressive strain, providing continuous heat conduct paths in both in-plane and through-plane directions at compacted status. PI coated on the cell walls of rGO aerogel by in-situ polymerization acts as the solder to “weld” the rGO sheets up through the covalent bonding. After the graphitization treatment, PI transforms into the graphitic structure and integrates with rGO sheets, providing a channel for phonon transmission and leading an enhancement of thermal transport performances.

## 2. Materials and Methods

### 2.1. Preparation of the Compressible Reduced Graphene Oxide (rGO) Aerogel

GO was prepared by the modified Hummers method. Then the partially reduced GO hydrogel was synthesized by heating the GO aqueous solution (4 mg/mL) with L-ascorbic acid (mass ratio, L-ascorbic acid:GO = 2:1) for 30 min at 90 °C. Then the hydrogel was treated by the freeze-thaw process at −18 °C and at 25 °C. The further reduction process was carried out for 5 h at 90 °C using the initial reductant. Finally, the compressible rGO aerogel was obtained by drying the rGO hydrogel at 60 °C for 8 h.

### 2.2. Preparation of the Compressible g-rGO/PI Aerogel

Polyamic acid (PAA) as PI precursor was synthesized from 0.1 mol of 3,3′,4,4′-biphenyl tetracarboxylic diandhydride and 0.1 mol of ρ-phenylenediamine mixed in 20 mL dimethylacetamide (DMAC) solvent. A polycondensation process was carried out at −5 °C for 5 h and the PAA solution was obtained. The PAA solution with PAA content of 5 wt% was obtained by diluting the solution in DMAC. Then the rGO aerogel was dipped into the PAA solution for 1 h in a vacuum condition to homogenously coat PAA on the rGO cell walls. Then the aerogel were immersed in anhydrous tert-butyl alcohol (TBA) for 48 h to exchange DMAC, ensuring the uniform coating of PAA on the cell walls and preventing the shrinkage of the cell walls during the solvent removal. After a freeze-drying process, the TBA was also removed.

To obtain the rGO/PI aerogel, the as-prepared rGO/PAA aerogel was subjected to thermal imidization at 250 °C for 1 h with protection of Ar atmosphere. Then the aerogel was carbonized at 1000 °C under Ar atmosphere for 1 h. Finally, the g-rGO/PI aerogel was obtained by the graphitization process at 2800 °C for 2 h in the graphitization furnace under Ar atmosphere. For comparison, g-rGO aerogel was also prepared to discuss the effect of g-PI on the thermal transport properties of the aerogel.

### 2.3. Characterizations

The microstructure of the aerogels was observed by a field-emission scanning electron microscope (SEM, Hitachi S4800, JEOL, Tokyo, Japan) equipped with energy-dispersive spectroscopy (EDS). Fourier transform infrared spectroscopy (FTIR) spectra of the aerogels were collected using a Fourier transform infrared (FTIR) spectrometer (Nicolet5700, Thermo, Waltham, MA, USA). Thermogravimetric analysis (TGA) was performed via a thermogravimetric analyzer (TGA, SDTQ600, TA Instruments, New Castle, DE, USA) in nitrogen atmosphere to determine the mass changes of aerogels with increasing temperature. Raman spectroscopy was performed using a LabRAM spectrometer (LabRAM HR Evolution, Horiba, Paris, France) with a laser of 532 nm. X-ray photoelectron spectroscopy (XPS) spectra were collected using a Physical Electronics PHI-5300 spectrometer (Physical Electronics, Minneapolis, MN, USA) with a monochromatic Mg Ka radiation at a voltage of 14 kV and a power of 250 W. Compressive stress/strain tests were performed using the Instron 5843 single-column system (Instron, Norwood, MA, USA) equipped with 1 KN load cell. The strain rate was 100 mm min^−1^.

### 2.4. Measurement of Thermal Transport Properties

The thermal transport properties of the aerogel TIMs were measured by the equipment designed in accordance with the standard of ASTM D5740. As shown in Figure 1a, for measuring the thermal transport properties in the through-plane direction, the aerogels were inserted between two Cu bars and pressed into thin films at 95% compressive strain. For testing the thermal transport properties in the in-plane direction, the compressed aerogels rotated from horizontal position to vertical position while maintaining the compressive strain of 95% with the help of the horizontal pressure (Figure 1b). The thermal interface resistance (*R_TIM_*) was calculated from the data obtained by the thermocouples located in the Cu bars. Thermal contact resistance (*R_c_*) and TC (*k*) were calculated by the simultaneous linear equations of Equation (1) by varying the dimensions (*l*) of samples [28]. The value of *l* was the thickness of the compressed aerogels for measuring the thermal transport properties along the through-plane direction. For measuring the thermal transport properties along the in-plane direction, the value of *l* was the length of the compressed aerogels.
(1)$$R_{TIM}=2R_{C}+\frac{l}{k}$$

## 3. Results

Figure 2 displays the formation process of the g-rGO/PI aerogel. At the pre-reduction stage, the increasing hydrophobicity of pre-reduced GO and π-π conjugated structure lead to the self-assembling of weakly cross-linked hydrogel. In the subsequent freeze-recast stage, the growth of ice crystals breaks through the weakly cross-linked network and pushes the pre-reduced GO sheets into the neighboring ice crystals to form the closely packed cell walls. During the further reduction and drying process, the compressible rGO aerogel with oriented cellular structure is obtained. Subsequently, the PI layers coat the cell walls of rGO aerogel by the in situ polymerization process. Finally, the PI films are transformed into g-PI layers coated on the framework after the graphitization process. By contrast with the cell walls consisting of stacked rGO sheets with abundant boundaries, the g-PI “welds” the rGO sheets on the surface of cell walls forming the integrated thermal transport paths in the aerogel, which can minimize the phonon scattering and improve the heat transfer efficient.

The micro-morphology of the compressible rGO aerogel is shown in Figure 3. It can be found that the as-prepared rGO aerogel possesses the oriented cellular architecture at the cross-section view (Figure 3a) and the vertical-section view (Figure 3b). The cell dimension in the aerogel is in the order of hundreds of micrometers (Figure 3c). The cell walls (Figure 3d) are composed of stacked rGO sheets forming a 3D interconnected network. The micro-morphology of the graphene aerogel in this work is similar to that in previous literature [32,33].

Figure 4a shows the morphology of the rGO/PAA aerogel. It can be found that after the introduction of PAA, the porous cellular structure of aerogel can be maintained well. The coating morphology is determined by the uniform distribution of a polymer precursor on the rGO cell walls (Figure 4b). Larger ripples appear on the cell walls that are different from the cell wall surface of pure rGO aerogel (Figure 3d), implying the in situ polymerization of PAA. The EDS mapping results of C, N elements also confirm the uniform distribution of the PAA on cell walls (Figure 4c,d). As shown in Figure 4e, after the imidization and subsequent graphitization, the oriented cellular structure of the aerogel is preserved well. The rough surface of the cell walls in rGO/PAA aerogel transforms into the smooth surface with unique wrinkle morphology of graphene (Figure 4f). During the graphitization treatment, the C atoms in rGO sheets would be rearranged, contributing to the repair of defects and a higher crystallinity. The coated PI layer as an additional carbon source turns into graphitic structure during the graphitization treatment, connecting and repairing the rGO plates. Graphitized PI layers are not only coated on the cell walls but also welds and connects the adjacent rGO sheets to make the cell walls more integrated. Such a structure with less interface can minimize the phonon scattering, thereby enhancing the heat transfer performance.

The FTIR spectroscopy is performed as shown in Appendix A to investigate the structure variation of the as-prepared samples. The characteristic bands of GO at 1070 cm^−1^, 1452 cm^−1^, 1637 cm^−1^ and 1720 cm^−1^ are attributed to the stretching vibration of C–O–C, C–OH, C=O and C=C groups, respectively, indicating the presence of C–O–C and –COOH groups on GO sheets [34,35]. For the rGO aerogel, the peak intensities corresponding to the oxygen-containing groups decrease, confirming the reduction of GO. After in-situ polymerization process and imidization treatment process, the rGO/PI aerogel shows a new peak at 1562 cm^−1^ contributed to the C–N–C stretching in PI [29,30,36]. After the graphitization treatment, the peaks related to nitrogen and oxygen containing functional groups that almost disappeared in the spectrum of g-rGO/PI aerogel. TGA was performed to quantify the reduction degree and thermal stability of the aerogels. As shown in Appendix A, the weight loss of rGO aerogel is about 22.6%, which represents the amount of residual oxygen-containing groups. For the g-rGO/PI aerogel, the weight loss is only 2.9%, indicating the almost complete elimination of nitrogen and oxygen containing groups and excellent thermal stability of g-rGO/PI aerogel after the graphitization [29,37,38].

The valance states and the chemical bonding of the elements in the aerogels were characterized by XPS in Figure 5a–c. The C 1s spectrum of rGO aerogel is deconvoluted into three peaks corresponding to C=C/C–C (~284.4 eV), C–O–C (~286.4 eV) and O–C=O (~289.1 eV) (Figure 5a) [10,12]. After the in-situ polymerization process, an additional peak at ~285.8 eV corresponding to the C–N bond can be observed, which demonstrates the existence of PAA in the aerogel (Figure 5b) [34,35]. As shown in Figure 5c, the N 1s spectrum of G/PI aerogel, can be divided into three peaks at 398.9 eV, 400.2 eV and 401.8 eV, arising from C–N–C, –CONH and –NH–, respectively [7,10]. The presence of -CONH indicates that the rGO sheets are “welded” by PI through the formation of -CONH bindings with terminal amine groups of PI and residual –COOH or C–O–C groups during the imidization [29,30]. The deoxygenation from rGO aerogel to g-rGO/PI aerogel after the graphitization process at 2800 °C was also investigated by XPS spectra. As shown in Appendix A, the O1s peak of g-rGO/PI aerogel became negligible, and the C/O atomic ratio of pure rGO aerogel is 4.38, improved remarkably to 72.47 for g-rGO/PI aerogel, confirming the almost complete elimination of oxygen containing groups after the graphitization. The pronounced and sharp C=C/C–C peak of g-rGO/PI aerogel indicates the significant restoration of sp^2^ bonded carbon lattice structure in the aerogel (Appendix A).

Raman spectroscopy was also performed to investigate the structural integrity of g-rGO aerogel and g-rGO/PI aerogel. As shown in Figure 5d, the D-band peak (1300 cm^−1^) of g-rGO/PI aerogel corresponding to the disordered amorphous carbon C–C bonds at the edge boundaries or defects have almost disappeared [39]. The G-band peaks (1580 cm^−1^) corresponding to the first-order scattering of E_2g_ mode of sp^2^-hybridized carbon atoms are narrow and sharp [40]. The value of *I_D_/I_G_* ratio of g-rGO aerogel and g-rGO/PI aerogel are 0.10 and 0.05, respectively. The decrease of *I_D_/I_G_* ratio indicates that the rGO sheets are “welded” and the grain boundaries between rGO sheets are filled up by g-PI during the graphitization treatment.

As mentioned above, 3D graphene monoliths commonly possess very low TC values due to the high porosity. Thus, the porous TIMs should be highly compressible so they can be compacted to reduce the porosity. Figure 6 shows the morphology change of g-rGO/PI aerogel during the compress/recovery process. When the aerogel is at ~70% compressive strain, the void space shrinks significantly and the solid matrix becomes denser (Figure 6a,b). No structural collapse or buckling failure is found on the cellular structure. Figure 4c indicates that the cellular structure of the aerogel maintains well under compression. When the applied load is released, the g-rGO/PI aerogel can fully recover the original cellular architecture (Figure 6d). The excellent compressibility of the g-rGO/PI aerogel results from the regular cellular structure [21,22,23] and the tightly packed cell walls “welded” by g-PI [29,30]. The g-rGO/PI aerogel is particularly effective at tolerating large compression yet preventing structural damage. Thus, the continuous heat conduct paths in g-rGO/PI aerogel can be preserved well at compacted status.

The compression measurements were carried out to evaluate the compressibility of g-rGO/PI aerogel. As shown in Figure 6e, the g-rGO/PI aerogel can be squeezed into a thin film under manual compression. The aerogel can almost completely recover to the original shape once the external pressure is removed. This high compressibility of the g-rGO/PI aerogel is consistent with the SEM observation mentioned above (Figure 6a–d). The cyclic strain/stress curves of the g-rGO/PI aerogel are shown in Figure 6f. The recoverable compressive strain of the aerogel is as high as 95%, which is similar to the highest values reported previously [21,32]. The loading process of the g-rGO/PI aerogel shows the characteristic behavior of porous foam-like materials, with three distinct regions, including the elastic region, plateau region and densification region [21,22,23]. 

The thermal transport performances of the as-prepared aerogels was studied. For the g-rGO/PI aerogel without compression, in-plane TC and through-plane TC values were only 6.3 W m^−1^k^−1^ and 4.1 W m^−1^k^−1^, respectively (Figure 7a). When the g-rGO/PI aerogel is at 95% compressive strain, in-plane TC and through-plane TC values of the compacted aerogel reached 172.5 W m^−1^k^−1^ and 58.1 W m^−1^k^−1^, respectively (Figure 7b). The improvement in TC values of the compacted g-rGO/PI aerogel was attributed to the significant decrease of the porosity of the aerogel under high compressive strain. The g-rGO/PI aerogel could be simply modeled as porous conductive film. By treating the pores as randomly sized spheres, the relationship between TC values and porosity of the porous film can be analyzed with the Bruggeman assumption [41]:(2)$$\frac{k_{aerogel}}{k_{G}}={\left(1-f\right)}^{3/2}$$
where *f* is the porosity, *k_aerogel_* and *k_G_* is the TC of the aerogel and the solid graphene. It can be found that the TC of the aerogel can be improved significantly by decreasing the porosity. The compression effectively decreases the porosity of the g-rGO/PI aerogel and leads to densification. Air as a bad conductor of heat (TC of ~ 0.22 W m^−1^K^−1^ [1]) inside of rGO cells can be removed by the compression. The previous literatures also reported that the densification process could improve TC values of graphene-based TIMs remarkably [8,9,10,11]. By contrast with the vertical graphene monoliths and horizontal graphene papers possessing high TC only along one-dimensional direction, the g-rGO/PI aerogel can maintain the 3D continuous thermal transport paths at the compacted status due the high compressibility of the aerogels, which leads to the aerogels possessing high TC values along both through-plane and in-plane directions.

Good contact between mating surface and TIMs is also important for highly efficient heat transport at the interface [1,2,3]. Thus, the thermal contact resistance has been discussed. For measuring the thermal transport performances in the through-plane direction, the thermal contact resistances between g-rGO/PI aerogel and Cu bars decrease from 21.6 K mm^2^ W^−1^ (without compression) to 6.4 K mm^2^ W^−1^ (at 95% compressive strain), which is attributed to the aerogel matching to the mating surface better with the help of the loading pressure. Along the in-plane direction, the thermal contact resistances between g-rGO/PI aerogel and Cu bars are 22.4 K mm^2^ W^−1^ (without compression) and 21.8 K mm^2^ W^−1^ (at 95% compressive strain), respectively. The almost similar values result from no high pressure applied at the contact interface along the in-plane direction. Owing to the change of TC and thermal contact resistance, the thermal interface resistance of compacted g-rGO/PI aerogel decrease from 2784.5 K mm^2^ W^−1^ (without compression) to 159.9 K mm^2^ W^−1^ (at 95% compressive strain) along the in-plane direction, and decrease from 2003.9 K mm^2^ W^−1^ (without compression) to 21.4 K mm^2^ W^−1^ (at 95% compressive strain) along the through-plane direction. The improvement of TC values plays a dominant effect on the decrease of thermal interface resistances.

The effect of g-PI welding on the thermal transport performances of the compacted aerogel has been studied by comparing the TC values between compacted g-rGO aerogel and compacted g-rGO/PI aerogel. As shown in Figure 7b,c, the TC values of compacted g-rGO/PI aerogel along both in-plane and through-plane directions are higher than that of compacted g-rGO aerogel (in-plane TC = 146.1 W m^−1^K^−1^, through-plane TC = 41.3 W m^−1^K^−1^). The improvement in TC values of g-rGO/PI aerogel is attributed to the integration of rGO sheets in the surface of cell walls by g-PI welding. For the g-rGO aerogel, although the GO sheets are reduced and the defects of rGO sheets are repaired, the interval between rGO sheets cannot be filled. Thus, the boundaries and gaps between rGO sheets leads to the phonon transportation. After the introduction of PAA into rGO aerogel, the gaps between the adjacent rGO sheets in the surface of the cell walls can be filled up by the PAA polymer. Then carbonization and graphitization processes make PI layers transform into a graphitic structure and integrate with rGO sheets [29,30]. The existence of g-PI enlarges the grain size of rGO sheets and provides a channel for phonon transportation and leads to the improvement of thermal transport properties.

The comparison of TC values and TC anisotropy ratio (higher-TC/lower-TC) between the compacted g-rGO/PI aerogel and other carbon-based TIMs reported in previous literature is shown in Figure 7d. TC anisotropy ratio is an important parameter for comprehensively evaluating the thermal spreading ability and the thermal transport efficiency of TIMs [42,43]. In comparison with the graphene foam/polymer composites [18,44], although the compacted g-rGO/PI aerogel presents a lower TC anisotropy ratio, both the in-plane TC and through-plane TC values of compacted g-rGO/PI aerogel are much higher due to the all-carbon components. Graphite/CNT hybrids [45,46] with 3D interconnected structure have high in-plane TC and through TC values. Although the in-plane TC of the compacted g-rGO/PI aerogel is slightly lower than that of graphite/CNT hybrids, the through-plane TC of compacted g-rGO/PI aerogel is higher and the TC anisotropy ratio is much lower. The excellent thermal transport performances suggest the promising potential of the g-rGO/PI aerogel in current demanding thermal management systems.

## 4. Conclusions

The g-rGO/PI aerogel with integrated cell walls and oriented cellular structure was fabricated by the ice template method and “molecular welding” strategy as the compressible TIMs with both high in-plane TC through-plane TC values. The closely packed cell walls and the cellular structure give the g-rGO/PI aerogel high compressibility, which makes the aerogel able to maintain 3D thermal transport paths well even at 95% compressive strain. In addition, the PI coating as a solder “welded” the rGO sheets up through the covalent bonding and to fill up the voids in the cell wall surface, providing a channel for phonon transportation in the 3D network. The compacted g-rGO/PI aerogel possesses in-plane TC of 172.5 W m^−1^k^−1^ and through-plane TC of 58.1 W m^−1^k^−1^, which is superior to other carbon-based TIMs with high thermal transport performances only along the one-dimensional direction. Our finding provides the insight for the construction of compressible TIMs, which may satisfy the demanding thermal management of next-generation high-power electronics.

## Figures and Tables

**Figure 1 materials-14-02350-f001:**
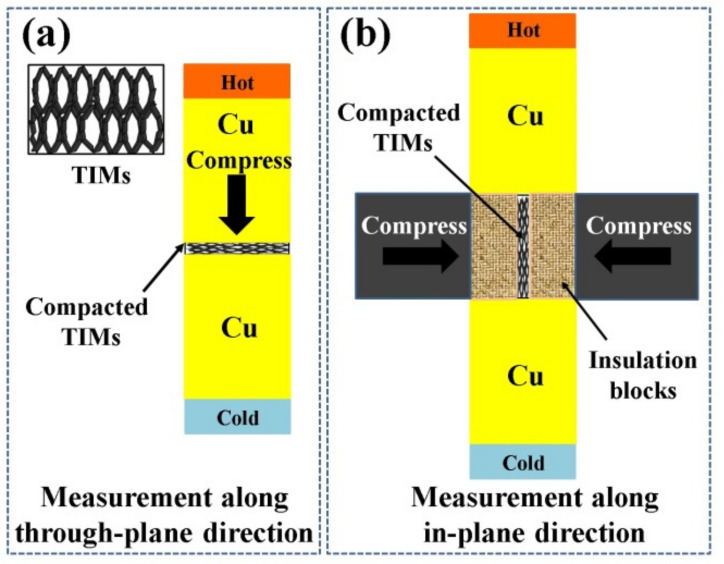
Illustration for testing the thermal transport properties of aerogel thermal interface materials (TIMs) along (**a**) through-plane direction and (**b**) in-plane direction, respectively.

**Figure 2 materials-14-02350-f002:**
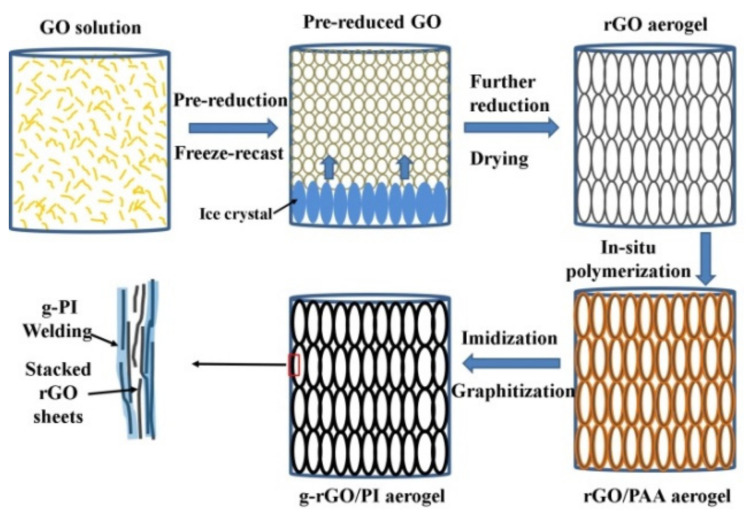
Illustration of the formation of graphitized reduced graphene oxide (g-rGO)/polyimide (PI) aerogel.

**Figure 3 materials-14-02350-f003:**
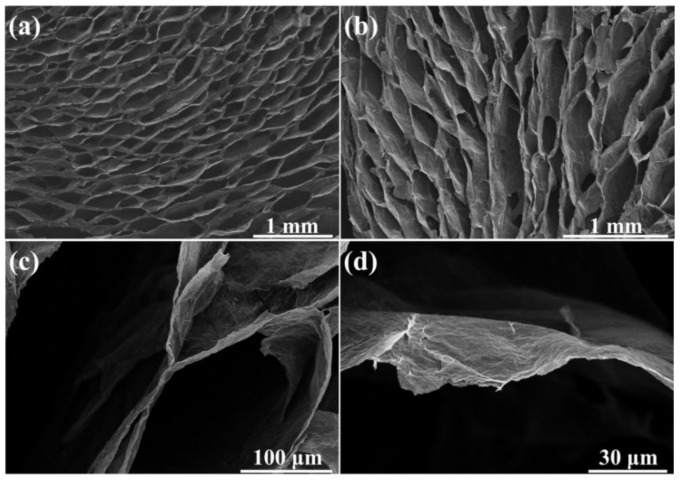
Scanning electron microscope (SEM) images of (**a**) cross-section and (**b**) vertical-section of the compressible rGO aerogel; (**c**,**d**) Cross-section of the compressible rGO aerogel at different magnifications.

**Figure 4 materials-14-02350-f004:**
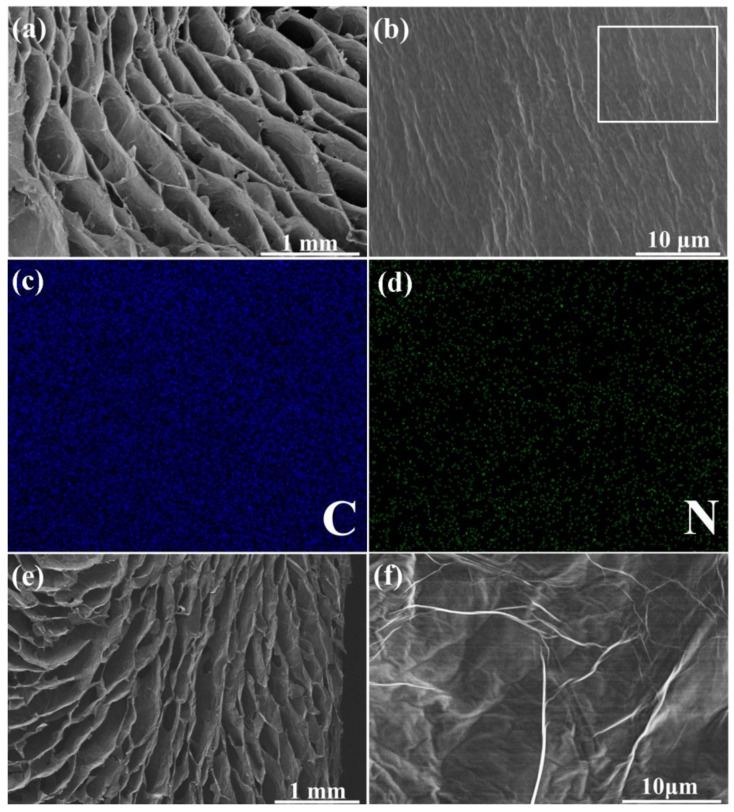
Scanning electron microscopy (SEM) images of (**a**,**b**) rGO/polyamic acid (PAA) aerogel; energy-dispersive spectroscopy (EDS) element mapping images of (**c**) C and (**d**) N in the selected area in (**b**); SEM images of (**e**,**f**) g-rGO/PI aerogel.

**Figure 5 materials-14-02350-f005:**
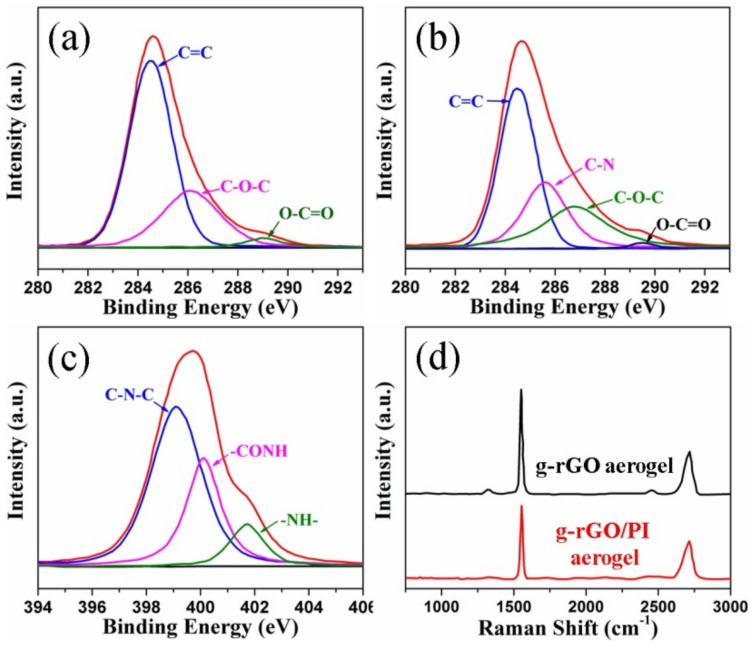
X-ray photoelectron spectroscopy (XPS) C1s spectra of (**a**) rGO aerogel and (**b**) rGO/PAA aerogel; (**c**) XPS N1s spectra of rGO/PI aerogel; (**d**) Raman spectra of g-rGO aerogel and g-rGO/PI aerogel.

**Figure 6 materials-14-02350-f006:**
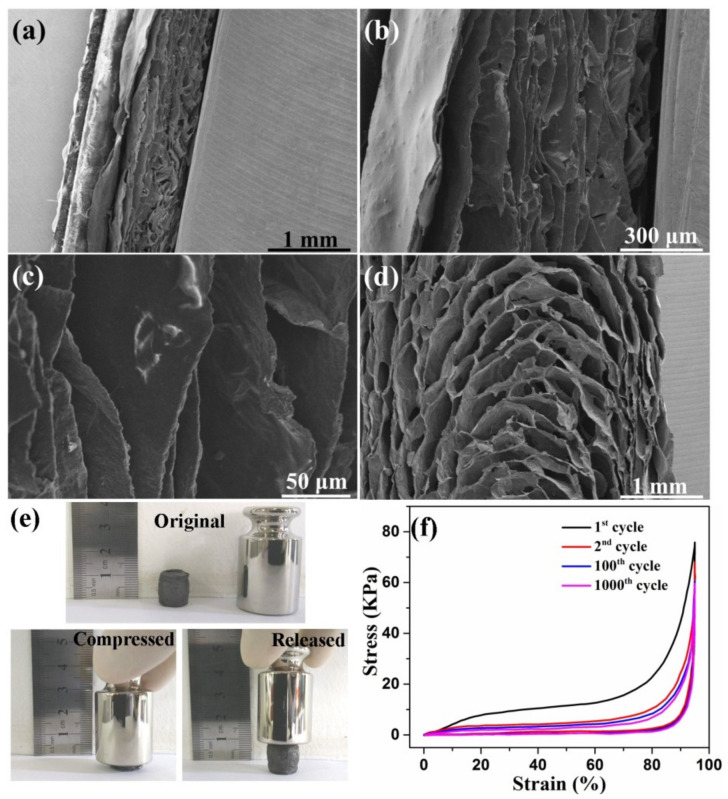
The cross-section SEM images of g-rGO/PI aerogel (**a**–**c**) at compression status and (**d**) at release status; (**e**) Digital photographs of the compress/recovery process of g-rGO/PI aerogel; (**f**) Stress/strain curves of the 1st, 2nd, 100th and 1000th cycles of g-rGO/PI aerogel.

**Figure 7 materials-14-02350-f007:**
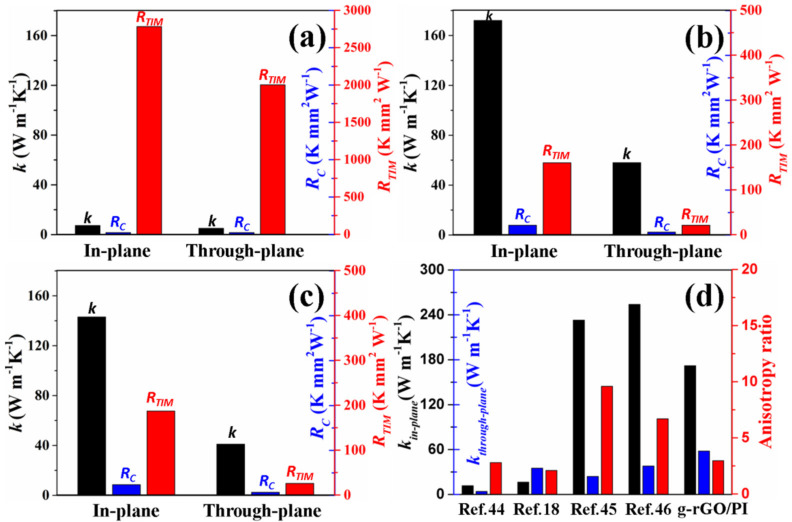
Thermal transport performances of (**a**) g-rGO/PI aerogel without compression, (**b**) g-rGO/PI aerogel at 95% compressive strain and (**c**) g-rGO aerogel at 95% compressive strain; (**d**) Comparison of thermal transport performances between the carbon-based TIMs in previous reports and the compacted g-rGO/PI aerogel.

## Data Availability

Not applicable.

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
