# Peer review of "Graphitized-rGO/Polyimide Aerogel as the Compressible Thermal Interface Material with Both High in-Plane and through-Plane Thermal Conductivities"

_materials, 2021, doi:10.3390/ma14092350_

Round 1

Reviewer 1 Report

Dear authors,

After reading the manuscript titled Graphitized-rGO/polyimide aerogel as the compressible thermal interface material with both high in-plane and through-plane thermal conductivities, I have the following comments to make for each section of this manuscript.

Abstract

I do not agree with the first sentence that says "Reduced graphene oxide (rGO) aerogels with three-dimensional (3D) interconnected network and continuous heat transport paths are the promising candidates for the advanced thermal interface materials (TIMs) with high thermal conductivity (TC) along both in-plane direction and through-plane direction" since the authors reveal the reasons why rGO has very low thermal conductivity, in the next sentence. Therefore, there is a contradiction in the abstract, from my point of view. In addition, the lines 20-22 are not well written, they should contain the definite article such as an example "The aerogel g-rGO / PI in compressive strain 20 of 95% has a thermal conductivity in the plane of 172.5 W m-1k-1 and a thermal conductivity in the plane of 58.1 W m-1k-1 , which is superior to other carbon-based TIMs previously reported."

It will probably be better to rewrite it.

Keywords

"compressible"? Fits better compressibility.

Introduction

Line 32: "…. thermal conductivity (TC) (3000~5000 W m-1K-1),…" I believe the authors wanted to say "…. thermal conductivity (TC) (3000 - 5000 W m-1K-1),…"

Line 44: "… TC values (35.5~680 W m-1K-1)…I believe the authors wanted to say "…. TC values (35.5 - 680 W m-1K-1)…"

Line 45: "… high anisotropic [17]." I believe the authors wanted to say "….high anisotropy [17]."

Line 49: "… literatures…" must be "…. Literature…"

Line 52: "… from of the..." must be "… from that of the…"

Besides these errors, in the penultimate paragraph, the authors only compared this work with their previous work ([25]). I suppose it would be interesting to mention other works. In addition, the relevance of using polyimide (PI) instead of carbon nanotubes should be well explored and supported.

Materials and Methods

Line 90: "… 3,3′,4,4′-Biphenyl 90 tetracarboxylic diandhydride…" must be "… 3,3′,4,4′-Biphenyl 90 tetracarboxylic dianhydride…"

During graphitization, high temperatures were used, such as 1000 ºC (line 102) and 3000 ºC (line 104). I am wondering what is the thermal stability of rGO at these high temperatures, as it normally decomposes between 600 - 800 ºC.

Line 105: "…also prepared discuss the effect of PI…" must be "…also prepared to discuss the effect of PI…"

Characterizations

The SEM / EDS are not a good methodology to analyze the surface morphology since we are talking about semiconducting materials.

I do believe that doing elemental analysis of the samples rGO and grGO-PI will let the authors to know the extent of reduction of GO and the extent of graphitization, respectively. Besides that, attenuated total reflectance (ATR) infrared absorbance spectra of the samples rGO and grGO-PI will demonstrate the change in structure after reduction and after graphitization with PI, respectively.  Due to my previous question on thermal stability, I advise the authors in doing thermogravimetric analysis (TGA) to further estimate the reduction degree of the samples and thermal stability.

Thermal characterization

Lines 127, 132 and 133: "…is…" must be "…was…"

Results

Line 158-159: the authors said: "… by stacked rGO sheets as the stretched ligaments are joined at nods and junctions, forming a 3D interconnected network". I ask where the authors see these events in figure 3(d)? I am not able to see them…

The results are well discussed; most suffer from the lack of bibliographic support. Any discussion must be well-founded and even compared with previous studies. I believe that there are good references that support what the authors describe.

Conclusions

They are fine.

References

I think there is a lack of some good reviews on the subject in the literature, such as the link 10.1109 / JPROC.2006.879796 (828 citations) and the link 10.1080/09506608.2017.1296605 (98 citations but is more recent).

Reviewer 2 Report

The authors experimentally study the thermal transport performances of the g-rGO/PI aerogel with and without compressive strain as high as 95%. They prepared rGO aerogel that was treated with Polyamic acid and subsequently thermalized and carbonized it to create graphitized-rGO/PI aerogel which was examined to confirm the effect of the PI structure on the thermal transport properties for in-plane and through-plane directions relative to the compressive strain applied. All stages of preparation and SEM characterization of the aerogel’s cell structure in the order of 100 μm  are thoroughly described. The graphitized Pi welds rGO sheets [4-5 layers] in the cell’s wall surface creating a thermal transport path between cells. However, the contact area of such welded sheets shown in Fig.3-4 between cells is not evaluated from the given data and can be only guessed. The graphitized Pi surface of the cells (300 - 100 μm) forms wrinkles on an order of 10 μm as seen from Fig. 4f. The chemical bonding in aerogel is confirmed by results of XPS and Raman spectra that indicate the formation of crystalline structure between rGO sheets in g-PI presence. Without compression, the thermal conductivity of the aerogel is low, 6.3 & 4.1 W m-1k-1, in both dimensions and should relate to the contact area between cell walls and size of the cells because the acoustic phonon propagation' mean free path is measured in the few μm [D.L. Nika, E.P. Pokatilov, A.S. Askerov, and A.A. Balandin, Phonon thermal conduction in graphene: Role of Umklapp and edge roughness scattering, Phys. Rev. B (2009)], [Tianli Feng, Xiulin Ruan, Zhenqiang Ye, and Bingyang Cao, Spectral phonon mean free path and thermal conductivity accumulation in defected graphene: The effects of defect type and concentration, Phys. Rev. B 91, 224301 (2015)].

After the 95% compression, the measured thermal conductivity remarkably rises up to 15-20 times. To explain such changes, authors should consider that the cell surface wrinkles [~10 μm] at 95% compression [100  → 5 μm] create additional contact areas within cells with Van der Waals interaction. And probably, contact area and distance between such compressed contacts create an additional network of phonon paths for in-plane and through-plane propagation. If such intracell contact area during compression could be discussed or qualitatively evaluated or characterized, it would benefit the discussion of the thermal transport properties of the g-rGO/PI aerogel under compressive strain. 

Round 2

Reviewer 1 Report

Dear authors,

After reading your responses to the reviewer's comments, I have to add the following:
First of all, it is not polite to write a text in red.

Second, I still have comments to add to the authors' responses, and many are concerned to the new data, which was not well supported, again.

Third, there is English mistakes.
